# Attentive Transfer Entropy to Exploit Transient Emergence of Coupling Effect

**Xiaolei Ru**[1], **Xin-Ya Zhang**[1], **Zijia Liu**[1], **Jack Murdoch Moore**[1]*, **Gang Yan**[1]*

[1] School of Physical Science and Engineering,
National Key Laboratory of Autonomous Intelligent Unmanned Systems,
MOE Frontiers Science Center for Intelligent Autonomous Systems,
Tongji University, Shanghai, China
`{ruxl,xinyazhang,xwzliuzijia,jackmoore,gyan}@tongji.edu.cn`

## Abstract

We consider the problem of reconstructing coupled networks (e.g., biological neural networks) connecting large numbers of variables (e.g., nerve cells) for which state evolution is governed by dissipative dynamics consisting of strong self-drive which dominates the evolution and weak coupling-drive. The core difficulty is sparseness of coupling effect, which emerges with significant coupling force only momentarily and otherwise remains quiescent in time series (e.g., neuronal activity sequence). Here we propose an attention mechanism to guide the classifier to make inference focusing on the critical regions of time series data where coupling effect may manifest. Specifically, attention coefficients are assigned autonomously by artificial neural networks trained to maximise the Attentive Transfer Entropy (ATEn), which is a novel generalization of the iconic transfer entropy metric. Our results show that, without any prior knowledge of dynamics, ATEn explicitly identifies areas where the strength of coupling-drive is distinctly greater than zero. This innovation substantially improves reconstruction performance for both synthetic and real directed coupling networks using data generated by neuronal models widely used in neuroscience.

## 1 Introduction

In this work, our task is to infer coupling relationships between observed variables based on time series data and reconstruct the coupled network connecting large numbers of these variables (see Figure 1a). Assume the time series $x_{it}$ records the state evolution of variable $i$ governed by coupled dissipative dynamics, as represented by a general differential equation $\dot{x_{it}} = g(x_{it}) + \sum B_{ij} f(x_{it}, x_{jt})$, where $g$ and $f$ are self- and coupling functions respectively. The parent variable influences the evolution behavior of the child variable via the coupling function $f$. Note that these two functions are hidden and usually unknown for real systems. The asymmetric adjacency matrix $B$ represents the directional coupling relationship between variables. Hence, the goal is to infer matrix $B$ from observed time series $x_{it}$, $i = 1, 2, \ldots, N$ where $N$ is the number of variables in the system. If $B_{ij} = 1$, the variable $j$ is a coupling driver (parent) of variable $i$, otherwise it is zero.

In neural dynamics (e.g., biological neuronal systems observed via neuronal activity sequences), the coupling effect is usually too weak to be detected, making less applicable the classic unsupervised techniques used across multiple research communities to infer coupling relationships [1, 2, 3, 4, 5, 6, 7, 8, 9, 10]. This difficulty manifests in three aspects. First, the dynamics contains self-drive and coupling-drive. The strength of the coupling force $f(x_{it}, \cdot)$ is usually many orders of

---

*Corresponding author

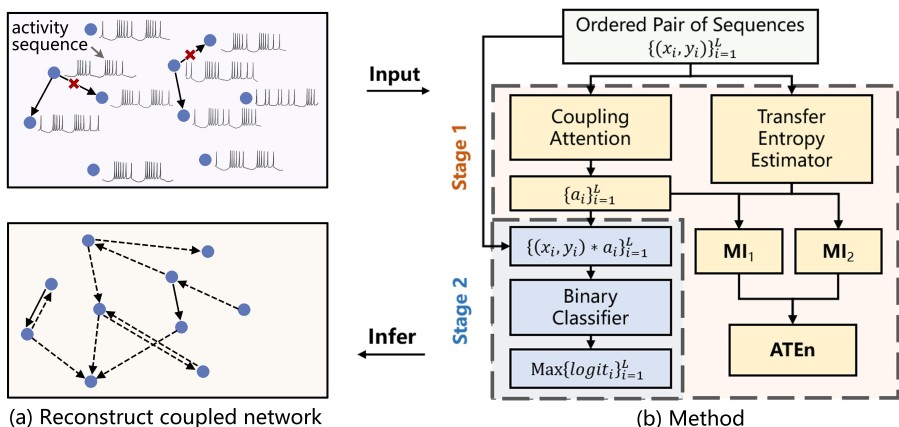

(a) Reconstruct coupled network          (b) Method

Figure 1: (a) Graphical representation of our task. With the activity sequences recording the state evolution of the variables and a small number of known relationships between variables (top panel: solid arrows with and without red crosses denote coupling and non-coupling respectively), our target is to infer the remained coupling relationships (bottom panel: dotted arrows) and reconstruct the coupled network; (b) Overall framework of our method. An input sample consists of two activity sequences on an ordered pair of variables. Stage 1: the neural network $g_\alpha$ assigns attention coefficients $\{a_i\}_{i=1}^{L}$ along sequences to maximize ATEn as Eq. 6. The neural networks $f_\theta$ and $f_\phi$ forming transfer entropy estimator estimate mutual information $MI_1$ and $MI_2$ (first and second terms in Eq. 4). Stage 2: Classifier $h_\eta$ infers the probability of coupling between variables.

magnitude smaller than self-drive $g(x_{it})$, and the latter dominates the evolution of variable state. Second, the behavior of the coupling-drive can be chaotic, unlike in linear models [11, 12]. The resulting unpredictability and variability of system state means that coupling force can be significant momentarily and otherwise almost vanish, as illustrated in Figure 4 (gray lines). This dilutes the information in time series that can be useful for inferring the coupling relationship. Third, in the heterogeneous networks common in applications, some variables are hubs coupled with many parent variables, among which it is difficult to distinguish individual drivers.

When coupling effects are weak and sparse, we do not observe clearly the principle of Granger Causality, whereby the parent variable can help to explain the future state change in its child variable [13]. Rather, when we train a model [5, 8] for prediction task on the neuronal activity sequences, the model only exploits the historical information of the child variable itself and that from parent variables is ignored. We posit that coupling-drive makes a negligible contribution to state evolution in the majority of samples of time series data. In other words, only in a small fraction of samples is the information of parent variables effective in predicting the evolution of child variables. Taking as an example the gradient algorithm to minimise the regression error over all samples $\sum_t (x_{it} - \hat{x}_{it})^2$, the adjustment of model parameters from the tiny samples corresponding to significant coupling force is negligible, but these are the only samples which could induce the model to exploit coupling effects in reducing regression error. Similarly, for transfer entropy [2], which measures the reduction in uncertainty which a potential parent variable provides to a potential child variable, there is no significant difference in measured value between ordered pairs of variables with and without coupling relation.

To overcome the difficulty, we introduce coupling attention to identify the moments when coupling effect emerges. We design an objective function, **A**ttentive **T**ransfer **En**tropy (**ATEn**), comprising a weighted generalisation of transfer entropy. In order to maximize **ATEn**, an artificial neural network is trained to autonomously allocate high attention coefficients $a_t$ at times $t$ where information of parent variables effectively reduces the future uncertainty of child variables, and ignores other positions by setting $a_t$ close to zero.

However, coupling attention also detects high transfer entropy regions produced by factors unrelated to coupling-drive, e.g., noise in empirical samples, which leads to spurious identification of spurious coupling effects between variables, which disturb the inference. To ameliorate this, we consider

utilising known relationships between variables, a small number of which could be inferred, e.g., by manual neuron tracing in connectome [14]. We leverage this limited number of known relationships for more effective detection of connectivity which advances in technology are making increasingly simple and cheap to observe [15]. Specifically, we add a binary classification model to perform more sophisticated inference under the guidance of coupling attention to focus on these critical regions and recognize different patterns between real and spurious coupling effect. We deal with this coupling relation inference task by way of small sample supervised learning. Although training and test data have a distribution shift in the setting of small samples, they arise through an identical underlying generation process. Thus, if the model provides an insight into the underlying dynamics – the coupling-drive in our task – then the understanding acquired from small samples can be effectively utilised in the test environment [16, 17, 18, 19]. The role of coupling attention is to help the classification model gain this insight. Our contributions are summarized as follows:

1. We draw on the coupling attention mechanism to identify the positions of time series at which coupling effect emerges and guiding a classification model to infer coupling relation by focusing on these critical positions. Without any prior knowledge of dynamics, this mechanism determines the areas where the coupling force is substantially different from zero.

2. By formulating Transfer Entropy as the difference between two types of mutual information, and based on the dual representation of Kullback-Liebler (KL) divergence, we design a differentiable metric, Attentive Transfer Entropy, as the objective function of the proposed coupling attention.

3. Our method significantly improves performance on synthetic and real directed coupling networks using the data generated by four well-known neural dynamic models, and the number of labels required is very small compared to the size of the coupled networks.

## 2 Background

### 2.1 Definition of Transfer Entropy

The transfer entropy, an information-theoretic measure, is able to detect information flow between time series $X$ and $Y$. Transfer Entropy measures the degree of non-symmetric dependence of $Y$ on $X$, defined as [2]:

$$TE(X \to Y) = \sum p\left(y_{t+1}, y_t^{(k)}, x_t^{(l)}\right) \log \frac{p\left(y_{t+1} \mid y_t^{(k)}, x_t^{(l)}\right)}{p\left(y_{t+1} \mid y_t^{(k)}\right)}, \tag{1}$$

where $x_t^{(l)} = (x_t, ..., x_{t-l+1})$ and $y_t^{(k)} = (y_t, ..., y_{t-k+1})$ and $k, l$ are lengths of memory. For an uncoupled system ($X$ and $Y$ independent) that can be approximated by a Markov process of order $k$, the conditional probability to find $Y$ in state $y_{t+1}$ at time $t+1$ satisfies $p\left(y_{t+1} \mid y_t^{(k)}, x_t^{(l)}\right) = p\left(y_{t+1} \mid y_t^{(k)}\right)$, and so $TE(X \to Y)$ vanishes.

### 2.2 Mutual Information Neural Estimation

The mutual information is equivalent to the KL divergence between the joint distribution $P_{XY}$ and the product of the marginal distributions $P_X \otimes P_Y$ [20, 21]. The KL divergence $D_{KL}$ admits the neural dual representation [22, 23]:

$$MI(X, Y) = D_{KL}\left(P_{XY} \| P_X, P_Y\right) \geq \sup_{\theta \in \Theta} E_{P_{XY}}\left[f_\theta\right] - \log\left(E_{P_X \otimes P_Y}\left[e^{f_\theta}\right]\right), \tag{2}$$

where the supremum is taken over parameter space $\Theta$ and $f_\theta$ is the family of functions parameterized by the neural network with parameters $\theta \in \Theta$. The mutual information neural estimator is strongly consistent and can approximate the actual value with arbitrary accuracy [23].

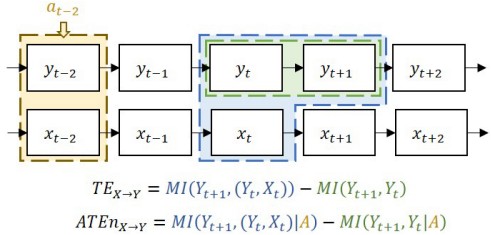

Figure 2: Visual interpretation of transfer entropy and its attentive version. Transfer Entropy is derived as the difference of two types of mutual information: $MI\left(Y_{t+1}, (Y_t, X_t)\right)$ (blue area) quantifies the reduction in uncertainty of future state $y_{t+1}$ from knowing current states $(y_t, x_t)$, and $MI\left(Y_{t+1}, Y_t\right)$ (green area) is same but only $y_t$ is known. Attention coefficients $a_t$ (yellow area) are assigned to each position of time series by coupling attention mechanism to maximize Attentive Transfer Entropy. For brevity, $k = l = 1$ here.

## 3 Method

### 3.1 Neural Estimator of Transfer Entropy

By the conditional Bayes formula and adding a marginal distribution of $Y$, we derive the transfer entropy as the difference between two types of mutual information. An intuitive description is provided in Figure 2, and the derivation is placed in Appendix A.

$$
TE(X \to Y)
$$

$$
= \sum p\left(y_{t+1}, y_t^{(k)}, x_t^{(l)}\right) \log \frac{p\left(y_{t+1}, y_t^{(k)}, x_t^{(l)}\right)}{p\left(y_r\right) p\left(y_t^{(k)}, x_t^{(l)}\right)} - \sum p\left(y_{t+1}, y_t^{(k)}\right) \log \frac{p\left(y_{t+1}, y_t^{(k)}\right)}{p\left(y_r\right) p\left(y_t^{(k)}\right)} \quad (3)
$$

$$
= MI\left(Y_{t+1}, \left(Y_t^{(k)}, X_t^{(l)}\right)\right) - MI\left(Y_{t+1}, Y_t^{(k)}\right). \quad (4)
$$

In these expressions, $y_r$ is sampled from $\mathbf{Y}$ randomly and independently of the time step $t$. The first term $MI\left(Y_{t+1}, \left(Y_t^{(k)}, X_t^{(l)}\right)\right)$ quantifies the reduction in the uncertainty of the future state $y_{t+1}$ from knowing the historical information $y_t^{(k)}$ and $x_t^{(l)}$. The second term $MI\left(Y_{t+1}, Y_t^{(k)}\right)$ is the reduction in uncertainty simply from knowing $y_t^{(k)}$. By connecting Eq. 4 and Eq. 2, we define a differentiable estimator of transfer entropy as:

$$
TENE(X \to Y) = \sup_\Theta E_{P\left(Y_{t+1}, Y_t^{(k)}, X_t^{(l)}\right)}\left[f_\theta\right] - \log\left(E_{P(Y_{t+1}) \otimes P\left(Y_t^{(k)}, X_t^{(l)}\right)}\left[e^{f_\theta}\right]\right)
$$

$$
- \sup_\Phi E_{P\left(Y_{t+1}, Y_t^{(k)}\right)}\left[f_\phi\right] - \log\left(E_{P(Y_{t+1}) \otimes P\left(Y_t^{(k)}\right)}\left[e^{f_\phi}\right]\right). \quad (5)
$$

Transfer entropy, and even mutual information, is difficult to compute [24], especially for high-dimensional or noisy data. In Appendix B, we offer a theoretical proof for the consistency and convergence properties of Transfer Entropy Neural Estimation, and examine its bias on a linear dynamic system where the true values of transfer entropy can be determined analytically.

### 3.2 Attentive Transfer Entropy

The main difficulty in our task is that the coupling effect in certain nonlinear dynamical systems is too weak to be recognized by classic techniques. We discuss the limitation of the iconic transfer entropy in detail that it works well when the three true distributions, i.e., one joint distribution and two conditional distributions in Eq. 1, can be estimated perfectly. However, sparse coupling effects are easily masked if the estimated probability density deviates even slightly from the real distribution. These momentary sources of evidence of coupling drive are like outliers in the total distribution of a time series dominated by self-drive. In order to make the transfer entropy provide a clear distinction between coupling and non-coupling pairs, we need to highlight the positions where $p\left(y_{t+1} \mid y_t^{(k)}, x_t^{(l)}\right) > p\left(y_{t+1} \mid y_t^{(k)}\right)$ and filter out other times by incorporating $a_t$ into Eq. 5, all

while avoiding the problem of distribution approximation. We do so by defining **ATEn** as:

$$ATEn(X \to Y) = \sum a_t \cdot p\left(y_{t+1}, y_t^{(k)}, x_t^{(l)}\right) \log \frac{p\left(y_{t+1} \mid y_t^{(k)}, x_t^{(l)}\right)}{p\left(y_{t+1} \mid y_t^{(k)}\right)}$$

$$= MI\left(Y_{t+1}, \left(Y_t^{(k)}, X_t^{(k)}\right) \mid A\right) - MI\left(Y_{t+1}, Y_t^{(k)} \mid A\right). \tag{6}$$

In this expression, $a_t \in [0, 1]$ is the attention coefficient at time step $t$ and the collection $A$ of attention coefficients is the attention series. Comparison of Eq.1 and Eq.6 reveals that the transfer entropy can be viewed as a simplified version of **ATEn** in which attention coefficients are uniformly set to one: $\forall t, a_t = 1$. Because each position has an equal contribution to estimation, the value of transfer entropy is dominated by the majority of positions where coupling effect is negligible, i.e., where $p\left(y_{t+1} \mid y_t^{(k)}, x_t^{(l)}\right) \approx p\left(y_{t+1} \mid y_t^{(k)}\right)$. Similarly to transfer entropy, **ATEn** is derived as the difference of two mutual informations, but **ATEn** incorporates the scheme of attention assignment. By connecting Eq. 6 and Eq. 2, we define a differentiable estimator of **ATEn** as:

$$ATEn(X \to Y) = \sup_{\Theta} \frac{1}{L} \sum a_t \cdot f_\theta\left(y_{t+1}, y_t^{(k)}, x_t^{(l)}\right) - \log\left(\frac{1}{L} \sum a_t \cdot e^{f_\theta\left(y_r, y_t^{(k)}, x_t^{(l)}\right)}\right)$$

$$- \sup_{\Phi} \frac{1}{L} \sum a_t \cdot f_\phi\left(y_{t+1}, y_t^{(k)}\right) - \log\left(\frac{1}{L} \sum a_t \cdot e^{f_\phi\left(y_r, y_t^{(k)}\right)}\right), \tag{7}$$

where $T$ is the total number of time steps and $L = T - \max(k, l)$. The expectation on the distribution of variables is adapted into the mean over time series.

## 3.3 Application of Coupling Attention

The overall framework of our model is presented in Figure 1b. In addition to two neural networks $f_\theta$ and $f_\phi$ for mutual information estimation, we employ another neural network $g_\alpha$ for coupling attention assignment. Rather than approximating distributions, the neural network $g_\alpha$ learns to maximize **ATEn** given by Eq. 7 via gradient descent. However, relying solely on coupling attention mechanism would lead to erroneous identification of spurious coupling effects as high transfer entropy regions which can also arise from non-coupling factors, e.g., noise in empirical samples. For more sophisticated inference, we augment our method with a binary classifier $h_\eta$ guided by coupling attention to focus on high transfer entropy regions and recognize different patterns between real and spurious coupling effect. The classifier takes as training samples a small number of ordered pairs of variables, among which labels for both coupling and non-coupling relationships are represented.

Then, the training process is divided into two independent stages: coupling attention learning and classification learning. The objectives in the first stage are:

$$\theta, \phi \leftarrow \underset{\theta, \phi \mid \alpha}{\operatorname{argmax}} \mathcal{L}_1 + \mathcal{L}_2 \qquad\qquad \alpha \leftarrow \underset{\alpha \mid \theta, \phi}{\operatorname{argmax}} \mathcal{L}_1 - \mathcal{L}_2 \tag{8}$$

where $\mathcal{L}_1, \mathcal{L}_2$ is the expectation of the first and second sup term of Eq.7 on training set respectively. We update $(f_\theta, f_\phi)$ and $g_\alpha$ alternately. A small learning rate is required to maintain training stability, otherwise the $g_\alpha$ may fall into a trivial solution where attention is almost zero throughout the time series. The objective in the second stage is:

$$\eta \leftarrow \underset{\eta \mid \alpha}{\operatorname{argmin}} \mathcal{L}_3, \tag{9}$$

where $\mathcal{L}_3$ is the binary cross entropy and the notation $\eta \mid \alpha$ indicates that coupling attention remains fixed during the second stage of training. The downstream classifier is sensitive to the upstream scheme of attention assignment[1]. Implement of our method is presented as Alg. 1 in Appendix C.

---

[1]When the ATEn reaches stability or convergence, the downstream classifier guided by it usually does not achieve optimal generalization in our experiments. Currently, we have not established a definitive criterion for determining the best stopping point of the first stage, at which the downstream classifier can achieve its optimal generalization. To address this issue, after every fixed number of epochs in the first stage, we retrain a new classifier with a few epochs. When the ATEn converges, we select one with optimal generalization on validation set from those classifiers and further train it until convergence.

## 4 Experiment

### 4.1 Setup

**Directed coupling networks.** For synthetic networks, we use Scale-Free (SF) graph model [25], which could generate networks with controllable structure characteristics. For real networks, we select six neurological connectivity datasets as presented in Table 1, each from a different species: Cat, Macaque, Mouse, C. elegans, Rat and Drosophila.

| Dataset | Region | #Nodes | #Edges | Mean degree |
|---------|--------|--------|--------|-------------|
| Cat | Cortex | 65 | 1139 | 17.5 |
| Macaque | Cortex | 242 | 4090 | 16.9 |
| Mouse | Cortex | 195 | 214 | 1.1 |
| C. elegans | Neural | 272 | 4451 | 16.4 |
| Rat | Cinerea | 503 | 30088 | 59.8 |
| Drosophila | Medulla | 1781 | 33641 | 18.9 |

Table 1: Statistical information of six real networks: dataset name, type of network, number of nodes, number of edges and mean degree $\langle k \rangle$. Details are provided in Appendix E.

**Dynamic models.** We use four dynamic models for neural activity simulation widely used in the field of neuroscience: Hindmarsh-Rose (HR), Izhikevich (Izh), Rulkov and FitzHugh-Nagumo (FHN). Dynamic equations are provided in Table 2, and segments of generated time series are represented in Figure 4.

| Name | Equations |
|------|-----------|
| HR | $\dot{p_i} = q_i - ap_i^3 + bp_i^2 - n + I_{\text{ext}} + \Gamma$ 
 $\dot{q_i} = c - dp_i^2 - q_i, \quad \dot{n_i} = r\left[s\left(p_i - p_0\right) - n_i\right]$ 
 $\Gamma = g_c\left(V_{\text{syn}} - p_i\right)\sum_{j=1}^N B_{ij}/(1 + \exp(-\lambda\left(p_j - \Theta_{\text{syn}}\right)))$ |
| Izh | $\dot{v_i} = 0.04v_i^2 + 5v_i + 140 - u_i + I + \Gamma$ 
 $\dot{u_i} = a(bv_i - u_i), \quad \Gamma = g_c\sum_{j=1}^N B_{ij}u_j$ |
| Rulkov | $\mathrm{F}_1(u_i, w_i) = \dfrac{\beta}{1 + u_i^2} + w_i + \Gamma\left(u_j\right)$ 
 $\mathrm{F}_2(u_i, w_i) = w_i - \nu u_i - \sigma$ 
 $\Gamma\left(u_j\right) = g_c\sum_{j=1}^N B_{ij}/\left(1 + \exp(\lambda\left(u_j - \Theta_s\right))\right)$ |
| FHN | $\dot{v} = a + bv + cv^2 + dv^3 - u + \Gamma$ 
 $\dot{u} = \varepsilon(ev - u), \quad \Gamma = -g_c\sum_{j=1}^N B_{ij}\left(v_j - v_i\right)$ |

Table 2: Equations of the four dynamical models considered. $B$ is the asymmetrical adjacency matrix of the coupling network, recording coupling relationships between nodes. $B_{ij} = 1$ if variable $i$ is the parent of variable $j$, otherwise $B_{ij} = 0$. In these expressions, $\Gamma$ describes the coupling-drive, while other terms represent self-drive. The detailed configuration of dynamical parameters is provided in Appendix D.

**Baselines.** We compare our method with eight baselines. Unsupervised learning: (1) Granger causality test (Ganger) [1]; (2) Transfer Entropy using the Kraskov-estimator (TE_Kraskov) [26], a standard method for TE calculation; (3) Convergent cross mapping (CCM) [3]; (4) Latent convergent cross mapping (Latent CCM) [27]; (5) PCMCI [6] and (6) PCMCI$^+$ [28] using partial correlation to quantify coupling strength. Unsupervised learning with validation set: (7) Transfer Entropy Neural Estimator (TENE), as in Eq. 5. The estimator is trained on the test set unsupervised and terminates upon achieving optimal performance on the validation set. Small sample learning: (8) Classifier with traditional attention mechanism [29], which was initially developed for computer vision tasks.

**Evaluation metrics.** We quantify the performance of methods by evaluating the metrics, specifically the area under the receiver operating characteristic curve (AUROC).

**Training details.** We employ a 4-layer convolutional neural network for model $g_\alpha$ and $h_\eta$, and a 5-layer fully-connected neural network for model $f_\theta$ and $f_\phi$. We use the ADAM [30] optimizer with initial learning rate of $10^{-3}$ for the classifier $h_\eta$ and $10^{-4}$ for the others. The learning rate decays exponentially by gamma = 0.999 per epoch. The batch size for stage 1 is 32 and for stage 2 is 10. The number of training epochs is 400. We run all experiments in this work on a local machine with two NVIDIA V100 32GB GPUs. See codes in Supplementary Materials for more details.

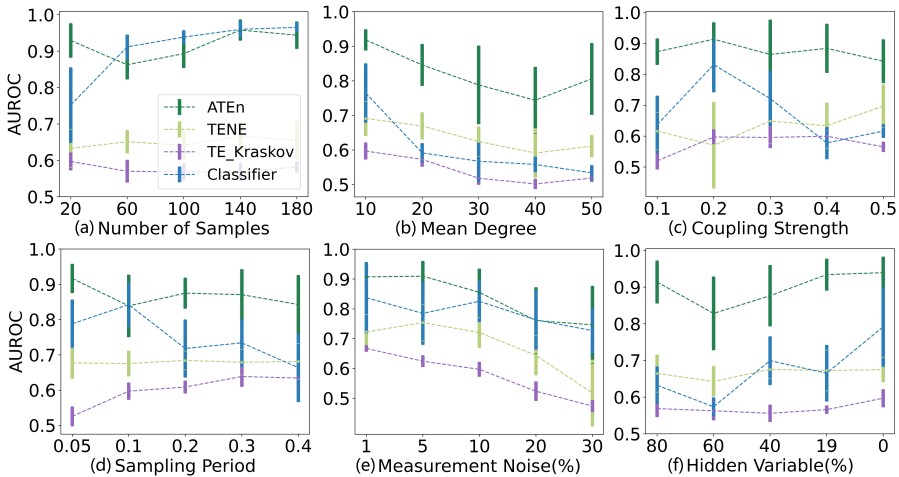

Figure 3: We conduct experiments of ablation study in six control environments including the number of samples using for supervised training, mean degree of networks (the higher mean degree results in more parents per child variable), coupling strength related to the extent of influence that parent variables have on child variables through the coupling drive, sampling frequency of activity sequence, measurement noise (Gaussian noise is added with mean zero and standard deviation of a certain proportion of the original sequence), and hidden variables (training or validation samples are randomly selected from the remaining observable variables, and test samples are randomly selected from the whole network). In the plots above, the default values of train samples is 20 (10/10 positive/negative samples), mean degree 10, coupling strength 0.2, sampling period 0.1, 10% measurement noise and 0% hidden variables, and each plot considers gradual variation of a single variable. Activity sequences are generated by the HR model on synthetic networks with 500 variables. The size of validation and test set is 100 and 400 respectively, with a uniform distribution of positive and negative samples as in the training set, which are randomly selected from all possible ordered pairs within the entire network.

## 4.2 Ablation study

Our method's innovation can be decomposed into three components: Firstly, a trainable transfer entropy calculator (TENE) is proposed; Secondly, TENE is combined with attention to detect region of high transfer entropy; Thirdly, the classifier is guided by coupling attention to focus on regions where the coupling effect may emerge. Here, we provide a detailed ablation study to determine which component is the key to performance improvement, and show results in Figure 3. We observe that Classifier alone (blue lines) exhibits the worse performances than our method (green lines) in most environments. This indicates that the spurious features learned by Classifier, despite quickly reaching low loss on the small training set, are less related to the properties of coupling effect and result in poor generalization capability on test set. When the size of train set increases, the performance of Classifier alone become comparable to that of **ATE**n (see Figure 3a). TENE (cyan lines) exhibits performance marginally superior to transfer entropy estimated using the standard Kraskov method (TE_Kraskov; purple lines). The **ATE**n (green lines) shows a substantial performance improvement compared with baselines, which indicates that the classifier with the guidance of coupling attention can extract features hidden in activity sequences that are closely related to coupling effect and result in superior generalization. This also demonstrates the importance of identifying and focusing on these critical regions of activity sequences. Moreover, our method is robust against changes to experimental conditions other than mean degree and measurement noise, a limitation we discuss in Sec. 5.

## 4.3 Insight into the coupling-drive of underlying dynamics

In Figure 4, we demonstrate the ability of coupling attention to catch the transient emergence of coupling effect. The gray lines in Figure 4 represent the change of coupling force from parent to child variable over time, and are generated by the coupling term $\Gamma$ in Table 2. The absolute value of the coupling force rises (the gray lines spike) at occasional moments when the behavior of a

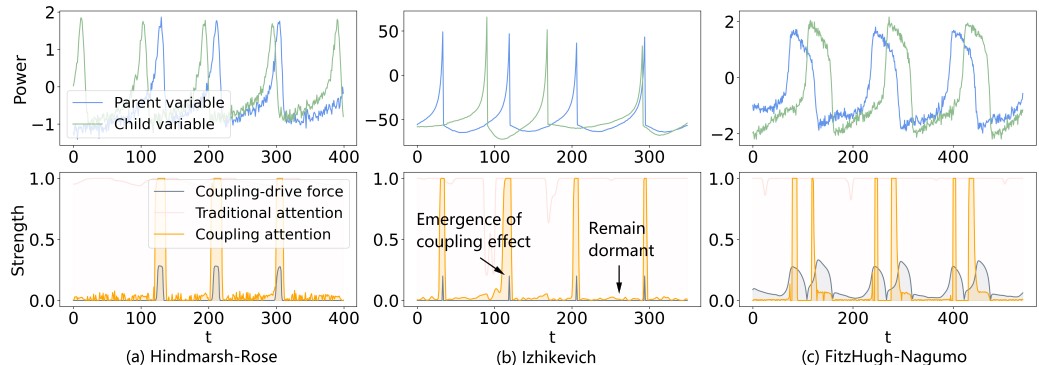

Figure 4: Insight into the coupling drive of neural dynamics. Top panel in each subplot: segment of two activity sequences on the ordered pair of variables with a coupling relationship (blue is parent and green is child). Bottom panel in each subplot: the absolute value of coupling force (gray line), the coefficient of traditional attention (light pink line) and coupling attention (orange line) assigned along the sequences. (a) HR; (b) Izh; (c) FHN. Time $t$ is index in the time series. In axis labels, "Power" is membrane potential, while "Strength" is membrane potential (attention coefficient) for the coupling force (attention). Results are chosen for clarity and are obtained when the training of stage 1 reaches stability.

parent variable substantially influences the evolution of its child variable, and remains almost zero at other times. The orange lines representing the coupling attention keep in step with the gray lines, indicating that the coupling attention mechanism recognizes the effect of coupling force in reducing the uncertainty of the child variable and pays attention to the areas where coupling force is significant. In contrast, the light pink lines in Figure 4 representing the traditional attention [29] remain close to their maximum value, indicating it is insensitive to changes in coupling force. This would lead its classifier to extract features from any part of the entire time series, rather than focusing on areas where coupling effects may emerge.

## 4.4 Performance on real networks

We test our method on six empirical neurological connectivity datasets using activity sequences data generated by four neuronal dynamic models. Compared with the baselines, our method usually substantially improves reconstruction performance on real coupled networks, as shown in Table 3. The performance of classical unsupervised methods, for which all positions in the time series are treated equally, is limited by the paucity of distinct coupling effects. In conclusion, our method slightly

Table 3: Performance comparison (model neurons coupled on real connectomes). Each point contains the mean and standard deviation of AUROC computed in ten experiments with randomly sampled training/validation/test set (20/100/1000) in **C. elegans** (left) and **Drosophila** (right) connectomes. Results on other three connectomes are shown in Appendix F.

| Dataset | C.elegans | | | | Drosophila | | | |
|---|---|---|---|---|---|---|---|---|
| Dynamics | HR | Izh | Rulkov | FHN | HR | Izh | Rulkov | FHN |
| Granger | 0.50±0.01 | 0.64±0.01 | 0.71±0.01 | 0.64±0.02 | 0.69±0.02 | 0.42±0.01 | 0.74±0.01 | 0.60±0.01 |
| TE Kraskov | 0.63±0.01 | 0.50±0.01 | 0.68±0.02 | 0.54±0.02 | 0.60±0.01 | 0.47±0.02 | 0.65±0.01 | 0.58±0.02 |
| CCM | 0.79±0.02 | 0.62±0.02 | 0.67±0.02 | 0.74±0.01 | 0.80±0.01 | 0.42±0.02 | 0.80±0.01 | 0.76±0.03 |
| Latent CCM | 0.78±0.01 | 0.55±0.04 | 0.53±0.04 | 0.68±0.01 | 0.81±0.02 | 0.47±0.02 | 0.76±0.01 | 0.73±0.01 |
| PCMCI | 0.53±0.01 | 0.53±0.02 | 0.66±0.01 | 0.57±0.02 | 0.51±0.01 | 0.51±0.01 | 0.51±0.01 | 0.51±0.01 |
| PCMCI$^{+}$ | 0.53±0.01 | 0.56±0.01 | 0.69±0.02 | 0.63±0.02 | 0.51±0.01 | 0.51±0.01 | 0.51±0.01 | 0.51±0.01 |
| Classifier | 0.75±0.06 | 0.65±0.05 | 0.72±0.06 | 0.65±0.07 | 0.85±0.05 | 0.65±0.06 | 0.83±0.09 | 0.72±0.05 |
| TENE | 0.65±0.06 | 0.43±0.03 | 0.65±0.08 | 0.47±0.06 | 0.54±0.07 | 0.51±0.03 | 0.54±0.11 | 0.41±0.06 |
| ATEn | **0.94**±0.06 | **0.66**±0.12 | **0.90**±0.07 | **0.91**±0.06 | **0.97**±0.03 | **0.74**±0.09 | **0.87**±0.06 | **0.83**±0.10 |

increases cost, due to the need for label collection, but obtains a substantial boost in performance compared with those unsupervised methods in this class of coupling network reconstruction tasks. We also observe that the reconstruction performances on identical dynamics across distinct connectomes are significant different (e.g., AUROC of **ATE**n on Izh across C.elegans in Table 3 and Rat in Appendix Table 5), indicating that the underlying network architecture plays an crucial role in shaping the behavior of coupling dynamics.

# 5    Limitations

Our methodology has limitations (i.e., cases for which performance improvement is less): 1. Dense networks, where a variable is coupled with many driving variables and substantial coupling forces can emerge from distinct parents at overlapping times, making individual drivers harder to distinguish (see green line in Figure 3b); 2. Intense noise, which makes the coupling attention mechanism falsely identify high transfer entropy regions. The downstream classifier then extracts spurious features, leading to the reduction of its generalization (see Figure 3e); 3. Strongly coupled systems, which are dominated by synchronization phenomena in which the dynamic behaviors of all variables are similar.

In addition, we assume the state evolution of variables in the coupled network is uniformly governed by a group of dynamic equations. However, if the evolution behaviors of variables in the system obey different underlying dynamics, it becomes critical to carefully select the small training samples, in which the patterns of coupled driving effects are representative. Failure to do so may result in the poor generalization of the classifier. That is our next research direction.

# 6    Related Work

**Coupled Network Reconstruction.** Several common methods of causal inference [31, 4, 6, 9] are based on conditional independence relations, but differ in the design of condition-selection strategies or choice of conditional independence test. Granger Causality [1] is extended to nonlinear dynamics by using neural networks to represent nonlinear casual relationships [8, 5]. Many methods of causal discovery assume that the coupled network is a directed acyclic graph. However, directed cyclic graphs are common in real systems. Conventional frameworks assume separability, i.e., that information about coupling drive are not contained in the parent variable itself. To address the non-separability issue, Convergent-cross mapping [3] and its variations [32, 27] measure the extent to which the historical record of child can reliably estimate states of the parent in reconstructed state space. However, sparse causal effect in neuronal dynamics, particularly in the presence of noise, may lead parent and child time series to appear statistically independent, so that their contribution to state estimation is hard to recognize.

**Mutual Information Estimation.** Ref. [23] built on a dual representation of KL divergence [22] to offer a parametric mutual information neural estimator (MINE) which is linearly scalable in dimension as well as sample size, and is also trainable and strongly consistent. They also discussed another version of MINE based on the $f$-divergence representation [33, 20]. Using the technique of Noise-Contrastive Estimation (NCE) [34], based on comparing target and randomly chosen negative samples, Van den Oord et al. [35] proposed InfoNCE loss, minimization of which maximizes a mutual information lower bound. An important application of this contrastive learning approach has been extracting high-level representations of different data modalities [36, 37, 38, 39]. In our work, we extend MINE for transfer entropy estimation.

**Attention Mechanisms.** The attention mechanisms identify key areas in the data by learning a set of weight distributions. Spatial-based attention [29, 40, 41] involves generating attention scores from spatial regions of feature maps, while channel-based attention [42, 43] optimises the representation of each channel. Self-attention [44, 45] encodes interactions among all input entities and cross-attention [46, 47, 48] introduces the interaction of two domains further. However, it is recognised that attention mechanisms need to be tailored to the specific problem at hand [49]. In our work, we tailor attention mechanism for coupling relationship inference by accommodating the selection of temporal regions that correspond to the transient emergence of coupling effect in neural activity sequences.

# 7 Discussion

The problem of reconstructing coupling networks from observational data is fundamental in multiple disciplines of science including neuroscience, since it is a prerequisite foundation for the research about structure analysis and behavior control in coupling networks. Especially, several countries have recently launched grand brain projects, and one important goal is to map the connectomes (i.e., directed links between neurons) of different species.

We draw on coupling attention to guide machine learning models to infer coupling relationships while focusing on the specific areas where casual effect may emerge. We show that this mechanism identifies weak coupling effects ignored by classical techniques, and helps machine learning models gain insight into the coupling dynamics underlying time series data.

## Acknowledgments and Disclosure of Funding

This work is supported by STI2030-Marjor Project 2021ZD0204500, National Natural Science Foundation of China (Grants No. T2225022, No. 12161141016, No. 12150410309), Shuguang Program of Shanghai Education Development Foundation and Shanghai Municipal Education Commission (Grant No. 22SG21), Shanghai Municipal Science and Technology Major Project (Grant No. 2021SHZDZX0100), Shanghai Municipal Commission of Science and Technology Project (Grant No. 19511132101), and the Fundamental Research Funds for the Central Universities.

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
