# OpenReview forum: "Attentive Transfer Entropy to Exploit Transient Emergence of Coupling Effect"
_NeurIPS.cc/2023/Conference — NeurIPS 2023 spotlight_

### Official Review · Reviewer_f3Wr · 2023-07-06

**Soundness:** 4 excellent
**Presentation:** 3 good
**Contribution:** 4 excellent
**Rating:** 8
**Confidence:** 4

**Summary:**

This article expands the concept of transfer entropy to include attentive transfer entropy. Implementing attentive transfer entropy on connectomes of realistic biological networks demonstrated that this extension encompasses the transient coupling effect within the connectivity constraints of realistic biological networks.

**Strengths:**

This paper provides convincing empirical evidence that the attentive transfer entropy is particularly useful in understanding coupling effects of realistic biological networks. They achieve this through first conducting ablation study on toy models of simple connectivity (full connectivity?) and comparing the proposed attentive transfer entropy with multiple commonly used baselines.



**Weaknesses:**

It is a bit confusing that the classifier with conventional attentional mechanism works almost as well as the ATEn in many scenarios (e.g., the Izh dynamics in C.elegans) and overall in Mouse connectome. In addition, it seems which dynamics being used to simulate the sequences may also influence the performance, but there was no discussion on that.


**Questions:**

1) The Classifier outperforms ATEn in mouse connectome across all dynamics. Does this mean ATEn may not work well in very sparse networks?
2) In many instances, the transfer entropy without attention mechanism performs significantly worse than both classifier and ATEn. Even the conventional attentional mechanism appears to be of great help. Is it possible to compare or discuss in some details the distinction between classifier and ATEn attention mechanisms in real-world networks?
3) The authors mentioned in a couple places that the train/validate/test split is 20/100/1000. How many samples are there in total? Is it possible to elaborate more training details?


**Limitations:**

This study employed predetermined dynamics, which is hardly realistic for functioning networks in the real world. The sensitivity to the choice of dynamics may hinder this method's potential utility for analyzing experimental results.

---

> ### Author Rebuttal · Authors · 2023-08-09
>
> We thank the referee for highlighting how we provided convincing empirical evidence that the attentive transfer entropy is particularly useful in understanding coupling effects of realistic biological networks.
>
> To Weakness:
>
> 1. Similarity of performance of different classifiers. Both the classifier with conventional attentional mechanism and ATEn do not work very well on the Izh dynamics in almost all networks since the strength of the transient couple force is weakest among all the dynamics used in our study. Please see Figure 4 for the gray lines visually depicting a temporal evolution of coupling force over time (Compared with left and right subplot, the spikes of the gray line in middle subplot are narrow and low). Thus, the couple effect of Izh dynamics is relatively hard to detect with either method, leading to similar performance.
>
> 2. The mouse connectome is a sparse network with low mean degree 1.1, indicating each child node has only one parent node approximatively. From the experimental results, the classifier indeed performs better than ATEn in mouse connectome. However, performance of classifier decreases quickly when the mean degree become large. Please see green (ATEn) and blue (Classifier) lines in Figure 3(b). ATEn already outperforms the classifier obviously with small supervised training samples when the mean degree of the synthetic network is 10.
>
> 3. The dynamics being used to simulate the sequences does indeed influence the performance of the method due to their different strength and frequency of the transient couple force, which ATEn relies upon.
>
> In our revision we will include a form of the above discussion.
>
>
> To Questions:
>
> 1. Performance of ATEn for sparse networks. The answer is presented in the reply to Weakness 2.
>
> 2. Distinction between classifier and ATEn attention mechanism in real-world networks. Thanks for the insightful comment. Biological neuronal networks function in an integrate-and-fire way, i.e., each neuron receives signals from its neighboring neurons, and it will fire only when the cumulative potential exceeds a threshold. Hence, the signal transferring from one neuron to another may not cause subsequential actions, which is the challenge of inferring directed link between neurons. Indeed, as shown in Figure 4 (pink curves), the classifier with conventional attention mechanism cannot capture such subtle fluctuations. Instead, our ATEn (Figure 4, orange curves) can do so. The main distinction is that conventional attention mechanism is to identify the regions of interest in activity series by maximizing cross-entropy while ATEn is to identify the important regions by maximizing transfer entropy. As we know, transfer entropy is a measure of causality between time series, able to capture the information directly transferring from one neuron to another. Such causation implies directionality of information flow, not just the differences between two series. ATEn is motivated by mimicking the transport of neuronal information, which might be the key reason why ATEn outperforms conventional attention mechanism.
>
> 3. Elaboration of training details. Taking a simple example, consider a directed network comprising 20 nodes and 100 edges. There are in total 20 * 19 = 380 ordered pairs of nodes, of which 100 have coupled relationships while the remaining 280 have non-coupling relationships. The information of five real networks used in our study is provided in Table 1. The train/validate/test split is 20/100/1000, meaning that we randomly sample 10/50/500 coupled and 10/50/500 non-coupling relationships respectively from all ordered pairs of the network. We will clarify these important details in our revision.
>
>
> To Limitations:
>
> Electric signals of biological neurons behave in a similar way of the several neuronal dynamical models, yet current experiment techniques usually measure neuronal activities in an indirect way, such as Ca+ signals. This leads to another problem: the measuring process itself is a nonlinear transformation of neuronal activity. Hence, the whole system should be modelled as: $\dot{x}_t=f(x_t)$ for neuronal dynamics, and $z_t=h(x_t)$ for the measuring process, where $h(*)$ represents a nonlinear function of measuring. To infer the underlying connectome from $z_t$ rather than $x_t$ is an outstanding problem and is also the future work we continue exploring now.

---

> > ### Comment · Reviewer_f3Wr · 2023-08-14
> > **Thank you for the clarification**
> >
> > We appreciate the authors' clarifications. Great work

---

### Official Review · Reviewer_wNf4 · 2023-07-07

**Soundness:** 3 good
**Presentation:** 3 good
**Contribution:** 3 good
**Rating:** 6
**Confidence:** 3

**Summary:**

This paper proposes "attentive transfer entropy" as a way to infer connectivity of real and synthetic coupling networks. The submission suggests extending a 'neural estimator of mutual information' to transfer entropy and adding "attention" to focus on moments in time where the influence is larger.


**Strengths:**

* higher performance than other methods this estimator is compared to.
* fewer data necessary than other methods this estimator is compared to.


**Weaknesses:**

* The introduction of attention appears to be ad hoc, a mathematical/information theoretic justification of the attention mechanisms seems to be missing.
* The methods seems a direct extension of the Mutual information neural estimator, the only technical innovation seems to be the attention mechanism which seems to lacks theoretic grounding.
* The whole theory (including the analytical toy example) considers discrete time dynamics but the methods is then applied to inference of couplings in systems of coupled differential equations. Something seems to be missing here to link the two descriptions (continous time and discrete time).

Minor issues:
* Table 1 is a bit confusing. The second column is "brain, brain cortex, neural, brain". This seems rather unspecific. What is the actual brain region in the cat an macaque data? Why don't you use a full connectome, e.g. of a fly (https://www.science.org/doi/10.1126/science.add9330)
* Weird notation in first equation. Why is t and index, usually ODEs are written as d x(t) /dt =f(x(t)) and not d x_t /dt=f(x_t)


**Questions:**

* Is the estimate of the transfer entropy assumed to be independent of the time-discretization? If yes, please provide evidence for that (e.g. show that changing dt of the numerical integration of the ODE does only have a very minor effect on the estimate of transfer entropy).
* How does the proposed method coping with confounders (ie. unobserved/unrecorded neurons)?
* In the single neuron models in the section 4 there is no noise added, so asymptotically, what is the source of entropy in this system? In other words: If the network dynamics is not chaotic, asymptotically the dynamics is expected to relax onto a fixed point or limit cycle. In both cases of fixed points and limit cycle, the true transfer entropy should be zero (observing Y very long is enough to predict it's future even without knowing X) so the full transfer entropy should be zero. What I am confused about is: Is this setup only quantifying some kind of 'transient' transfer entropy? Or is the approach relying on chaotic network dynamics? If yes, how does changing the intensity of network chaos affect the estimate? Or does the method actually implicitly rely on all nodes receiving independent noise? Is the 'numerical noise' enough? I am not expert on transfer entropy and I might be misunderstanding something here, but some clarification seems necessary.
* Why is noise added to the data sets? ("Gaussian measurement noise is added with mean zero and standard deviation 1%/10% that of the original time series for synthetic/real networks respectively.")
* What are the units of time in figure 3? Units of membrane time constant?
* How is this method dealing with identifiability issues, e.g. when different network structures can lead to the same dynamics?
* Could the proposed method be combined with causal interventions? E.g. can it tell me the neuron/synapse it has the biggest uncertainty about, so I could apply activity perturbations/causal interventions to infer if the neuron/synapse is connected or not?
* Why is the mean connectivity in the mouse data 1.1 according to table 1? It seems that compared to a real brain, possibly not all existing connections are in the data set. How does this change your conclusions? In other words: Real biological cortical neurons have a mean degree more arround 10^3 to 10^4 (see e.g. Potjans, Diesmann 2014), would your method still work for neural networks with realistic degree?

**Limitations:**

Limitations (mentioned above) are already addressed in the paper.

---

> ### Author Rebuttal · Authors · 2023-08-09
>
> We thank the referee for noting the strengths of our method: it provides higher performance and requires fewer data.
>
> To weakness and issues:
>
> 1. Discrete and continuous dynamics: In the analytical example and in the writing, for convenience, we used subscripts $t$ and $t+1$ because both experiment and simulation activity time series contain discrete data points. They actually denote $t + \delta t$ or $t + 2\delta t$ where $\delta t$ represents the time interval of each step. We showed the values of simulation time step $\delta t$ in Supplementary Section D.5.
>
> 2. Fruit-fly connectome: The paper about larva fly recently published in \it{Science} did not directly offer the connectome data and extracting it from their raw data is time consuming. Fortunately, we extract a connectome of the fruitfly Drosophila medulla from another data (Ref. [1] below), which has 1781 nodes and 33641 edges. The new results also confirm that our method outperforms the baselines, as shown in the table below. We will add this into the revision.
>
>                       HR          Izh        Rulkov         FHN
>        Granger     0.69±0.02   0.42±0.01    0.74±0.01    0.60±0.01
>        TE kraskov  0.60±0.01   0.47±0.02    0.65±0.01    0.58±0.02
>        CCM         0.80±0.01   0.42±0.02    0.80±0.01    0.76±0.03
>        Latent CCM  0.81±0.02   0.47±0.02    0.76±0.01    0.73±0.01
>        PCMCI       0.51±0.01   0.51±0.01    0.52±0.01    0.51±0.01
>        Classifier  0.85±0.05   0.65±0.06    0.83±0.06    0.72±0.05
>        TENE        0.54±0.07   0.51±0.03    0.54±0.11    0.41±0.06
>        ATEn        0.97±0.03   0.74±0.09    0.87±0.09    0.83±0.10
>
>    [1] L. K. Scheffer, et al. A connectome and analysis of the adult Drosophila central brain, eLife, 9: e57443 (2020).
>
> 3. Thanks. In the first equation, we personally find the notation $\dot{x}_t=f(x_t)$ more readable and less intimidating than the notation $\dot{x}(t) =f(x(t))$, especially taking into consideration the next several math equations (otherwise, too many parentheses). We feel that the notation will not induce ambiguity and hope it is okay to continue using our preferred notation.
>
> To Questions:
>
> 1. Impact of sampling frequency. We checked the effect of sampling frequency on the performance based on the estimates of transfer entropy, please see Figure 3(d). For our work, whether the estimates of transfer entropy themselves are precise is less important than whether estimates of transfer entropy between coupled variables are larger than estimates of transfer entropy between uncoupled variables.
>
> 2. Hidden nodes. We investigated the effect of hidden nodes on performance of the proposed method, please see Figure 3(f) which showing that the presence of hidden variables does not significantly affect performance. We consider an important reason is our consideration of the specific case of neuron dynamics in which the coupling-drive force is usually weak while self-drive force is strong. Another factor might be the fact that our inference is based on pairwise transfer entropy rather than a conditional transfer entropy which would require time series from all parent variables of each neuron.
>
> 3. Chaotic dynamics. So far, we considered chaotic dynamics because this work focuses on neuronal dynamics which are known chaotic. We thank the Reviewer for this insightful comment which is helpful for us to further extend our work to more general dynamical systems. We will acknowledge this by adding to the limitations “Our method may not apply to systems at fixed points or following limit cycles, for which the transfer entropy is zero.”
>
> 4. Measurement noise. We tested the effect of measurement noise intensity on the performance of the method in the section of Ablation study, please see Figure 3(e).
>
> 5. Units of t in Fig. 4. The units of the x-axis are sampling periods. In the caption to Fig. 4 in our revision, we will clarify this by changing the sentence “Time t is index in the time series” to “Time t is index in the time series and its unit is the sampling period.”.
>
> 6. Identifiability issues. Luckily, we would expect cases in which different structures give rise to exactly the same time series to be very rare for the dynamical systems which we consider.
>
> 7. The uncertainty is contingent upon the inferred probability of a causal link, $p \in [0,1]$, which is derived from our methodology. When $p$ approaches 1 or 0, the method exhibits high certainty regarding a connection. Conversely, when $p$ approximates 0.5, the method experiences maximal uncertainty. Nevertheless, we remain uncertain about the optimal locations for intervention experiments as this may rely on the relative value and difficulty associated with establishing or refuting a link. In our revision we will acknowledge this valuable idea by adding to the Conclusion: “In the future we hope to adapt our method to predict which pairs of neurons would be most valuable to have knowledge of whether or not a link exists, so that our method can be applied together with intervention experiments.”
>
> 8. Fruitfly and C. elegans neuron-level connectomes have moderate mean degrees. Connectome data of cat, macaque and mouse brains are not neuron-level but macroscopic connections between brain regions. They are all empirical data. The results of performance vs mean degree have been shown in Figure 3(b). Regarding large-degree neuron-level connectomes (i.e. such as dense networks with mean degree 10^3 to 10^4), we already discussed in the section “Limitations”: “Dense networks, where a variable is coupled with many driving variables and substantial coupling forces can emerge from distinct parents at overlapping times, making individual drivers harder to distinguish (see green line in Figure 3b)”.

---

> > ### Comment · Reviewer_wNf4 · 2023-08-14
> > **Acknowledgement of rebuttal**
> >
> > We thank the authors for the clarifications which have improved the study.
> > Some points are not fully clear to me:
> > * Regarding identifiability, the authors claim: "Luckily, we would expect cases in which different structures give rise to exactly the same time series to be very rare for the dynamical systems which we consider."
> > Can you support this claim with any evidence?
> > * Regarding the connectome: the adjacency matrix of the fly connectome can be downloaded here without the need to process the raw data: https://flywire.ai/
> > However, I can understand that this might be beyond the scope of this publication.
> > " Overall, I think the paper can be in its current form.

---

> > > ### Author Response · Authors · 2023-08-19
> > >
> > > 1. We thank the Reviewer for encouraging us to do theoretical analysis of identifiability, and we agree that it’s an important theoretical problem. We feel that, when the complexity of network topology increases, it will be less likely that another directed network can produce the same activity data and can have the same attentive transfer entropies as the ground-truth network does. However, it’s difficult to offer a theoretical proof for this assumption. Hence, in this paper we choose to validate our network inference method in various networks with different dynamics instead. We will work on this theoretical issue raised by the Reviewer in our next research.
> > >
> > > 2. Thanks for providing us a data source for a more comprehensive fruit-fly connectome and considering that our paper can be in the current form. We will make good use of this valuable dataset in our next research.

---

### Official Review · Reviewer_xCSe · 2023-07-07

**Soundness:** 3 good
**Presentation:** 3 good
**Contribution:** 3 good
**Rating:** 7
**Confidence:** 3

**Summary:**

The authors extend the MiNE method [23], of using a neural network to estimate mutual information, to estimating transfer entropy between two coupled time series. When the coupling is weak on average due to being intermittent, they propose a differentiable attention mechanism that learns to transiently gate the transfer entropy calculation to only these intermittent moments. They test their method to identify coupling in networks created using simple neuron models connected as per connectivity datasets.

**Strengths:**

-	The authors combine several new techniques to address an important issue of connectivity determination from neural recordings / time series.

**Weaknesses:**

-	The authors straightforwardly extend the theory of the neural estimation of mutual information to their neural estimation of transfer entropy without attention in Appendix A.. However, they do not say anything about their attention based method. Can it be shown that it is also somehow consistent / convergent?
-	It appears to me that all the ‘connectomes’ apart form the C.Elegans one are between macro brain regions. Yet these have been used to connect individual neuron models. Can the authors not use local connectivity of cortical neurons from say the Blue Brain project and deduce this connectivity?
-	Ultimately the method is only tested on simulated datasets even though the connectivity is based on macro-connectivity of animal brains. Numerous Ca-imaging or spike-train recordings of hundreds of neurons in various small regions of the brain are available. Yet the authors do not seem to have applied their method to these. Thus it calls into question whether their method is suitable for real-world data.


**Questions:**

Already stated in weaknesses.

Minor:
- L126: ‘adjusting $a_t$ in Eq. 5’ -> ‘adjusting $a_t$ in Eq. 6’ since Eq. 5 does not contain $a_t$.

---

> ### Author Rebuttal · Authors · 2023-08-09
>
> We thank the referee for noting that our paper addresses an important issue.
>
> To Weakness:
>
> 1. Consistency of ATEn. So far, we cannot theoretically demonstrate that ATEn is consistent in some way. At present, we have only experimentally verified that the attention model can converge to stability when the learning rate is small.
>
> 2. Neuron-level connectomes. This is a great point! We follow the Reviewer’s suggestion and complement our previous results with a neuron-level connectome from the fruitfly Drosophila medulla (Ref. [1] below). This connectome has 1781 nodes and 33641 edges. We confirm that our method outperforms the baselines using this new connectome, as shown in the following table.
>
>                       HR          Izh        Rulkov         FHN
>        Granger     0.69±0.02   0.42±0.01    0.74±0.01    0.60±0.01
>        TE kraskov  0.60±0.01   0.47±0.02    0.65±0.01    0.58±0.02
>        CCM         0.80±0.01   0.42±0.02    0.80±0.01    0.76±0.03
>        Latent CCM  0.81±0.02   0.47±0.02    0.76±0.01    0.73±0.01
>        PCMCI       0.51±0.01   0.51±0.01    0.52±0.01    0.51±0.01
>        Classifier  0.85±0.05   0.65±0.06    0.83±0.06    0.72±0.05
>        TENE        0.54±0.07   0.51±0.03    0.54±0.11    0.41±0.06
>        ATEn        0.97±0.03   0.74±0.09    0.87±0.09    0.83±0.10
>
>       [1] L. K. Scheffer, et al. A connectome and analysis of the adult Drosophila central brain, eLife, 9: e57443 (2020).
>
> 3. Experimental records of neuron activity. Yes, our ultimate goal is applying our method to infer real neuronal connectivity from the experimental recordings of neuronal activities. Before doing that, we wanted to test the method in various data of both neuronal activities and ground-truth connectivity. Unfortunately, neuroscience experimental data currently contain either neuronal activities or connectivity, and it is rare that they have both for a specific biological brain or region. Regarding Ca+imaging activity data, their sampling frequencies are usually low because multilayers of brain slices should be scanned for a specific 3D brain region. Hence, here we use real neuron-level connectome and use instead various neuronal dynamical models to generate activity data to validate our method. We thank the Reviewer for this insightful comment, and will discuss this (and spike-like discrete activity) in the section of limitations and future work in our revision.
>
>
> To Questions:
>
> Thanks. In our revision, we will change “adjusting a_{t} in Eq.5” to “incorporating a_{t} into Eq.5”.

---

> > ### Comment · Reviewer_xCSe · 2023-08-15
> > **read rebuttal**
> >
> > The authors have done well to add a neural-level connectome of the fruit fly to their tests to address point 2. It is good that they will clarify point 3 in their revised discussion, and I suggest they include point 1 as well in the discussion. With these revisions, I think the paper becomes better, and I can improve my rating to 7-accept  from 6-weak accept.

---

### Official Review · Reviewer_JWtX · 2023-07-30

**Soundness:** 2 fair
**Presentation:** 3 good
**Contribution:** 3 good
**Rating:** 7
**Confidence:** 3

**Summary:**

This paper proposes a novel method to reconstruct coupled networks from observational time series data, where the coupling effects between variables are sparse and weak. The authors propose a new attention mechanism called coupling attention is introduced to identify critical regions in time series where coupling effects momentarily emerge amidst dominant intrinsic dynamics. The authors design a differentiable objective function called Attentive Transfer Entropy (ATEn) to train the coupling attention model in an unsupervised, data-driven manner.



**Strengths:**

ATEn is well motivated and the use of Mutual Information Neural Estimation as a component to both estimate and optimize attention parameters makes sense.

The concept of using attention to identify sparse coupling effects in time series appears original for this reviewer and the method leverages the use of neural networks well. The method could likely be extended in the future to other neural architectures like transformers, therefore this work is a good initial foray into further work and improvements.

Text is well written and clear and the ablation experiments make sense.



**Weaknesses:**

There needs to be more clarity in the experimental setup:

Specifically, the way overfitting is prevented in the proposed ATEn method. Like Mutual Information Neural Estimation method, parameters are optimized in Transfer Entropy, which can result in perfect estimates of the mutual information (due to overfitting). How have the authors addressed this?

What are the inputs to fθ and fφ and how are they different from the inputs to the attention and classifier models? Is there a feature space beyond the scalar timeseries of the neurons?



**Questions:**

For the Hindmarsh-Rose experiments, why does it appear as though the child variable is leading the parent variable in the time-series in Figure 4?

What is the reason for making the attention and classifier convolutional networks while fθ and fφ are fully-connected networks?

For clarification, Figure 4 TENE refers to using Equation 5?



**Limitations:**

No obvious negative societal impacts. Refer to Weaknesses for constructive suggestions for improvement.

---

> ### Author Rebuttal · Authors · 2023-08-09
>
> We thank the referee for stating that ATEn is well motivated and for offering us constructive suggestions for improvement.
>
> To Weaknesses:
> 1. Preventing overfitting.
>
>    The coupling attention model is trained to maximize ATEn on the train samples. When the ATEn reaches stability (or converge to the optimal value), the downstream classifier guided by it usually does not achieve best generalization (performing well on the training set but poorly on the validation set) in our experiment. Currently, we have not established a definitive criterion for determining the optimal stopping point in training the attention model, at which point the downstream classifier guided by attention model can achieve its best generalization. We address this issue as follow: every $n$ training epochs of attention model, we retrain a new classifier with a few epochs and select the one with best generalization on validation set. After finishing the training of attention model, we further train the selected classifier until convergence.
>
>    In our revision, we will clarify this in Sec.4.1, “Setup”.
>
> 2. Inputs to $f_θ$ and $f_φ$.
>
>    We derive the transfer entropy to the difference between two mutual information (MI) as in Eq. 5; $f_θ$ estimates the first MI and $f_φ$ estimates the second MI. Taking a simple example, we have two neural activity sequences $X=[ x_{1}, x_{2}, x_{3}, x_{4}, x_{5}]$ and $Y =[ y_{1}, y_{2}, y_{3}, y_{4}, y_{5}]$ with the shape (5,) respectively, where subscripts represent timesteps.
>
>    $f_θ$ and $f_φ$ : Assuming $k=l=1$ in Equation 7, We can construct a pair of samples for $f_θ$ as $[x_{1}, y_{1}, y_{2}]$ and $[x_{1}, y_{1}, y_{\text{random}}]$ with the shape (3,) (Similarly, $[x_{2}, y_{2}, y_{3}]$ and $[x_{2}, y_{2}, y_{\text{random}}]$,… ), where $y_{\text{random}}$ represents it is sampled from sequences $Y$ randomly. We then put these pair of samples into $f_θ$ respectively. $f_θ$ outputs two values with the shape (1,) and is trained to maximize their difference to approximate the mutual information between $Y_{t+1}$ and $(Y_{t}, X_{t})$. We can construct another pair of samples for $f_φ$ as $[y_{1}, y_{2}]$ and $[y_{1}, y_{\text{random}}]$ with the shape (2,) (Similarly, $[y_{2}, y_{3}]$ and $[y_{2}, y_{\text{random}}]$,… ). We then put them into $f_φ$, which outputs two values with the shape (1,) and is trained to maximize their difference to approximate the mutual information between $Y_{t+1}$ and $Y_{t}$.
>
>    Attention model and classifier : We stack two sequences vertically and reshape them into an input sample $[[ [x_{1}], [x_{2}],[x_{3}],[x_{4}],[x_{5}] ], [ [y_{1}],[y_{2}],[y_{3}],[y_{4}],[y_{5}] ]]$ with the shape [1,2,5] (like [channel, height, width] for CNN). The attention model is fed by this sample and outputs an attention coefficient vector $[a_{1}, a_{2}, a_{3}, a_{4}, a_{5} ]$ with shape (5,), the same length as the activity sequences. We get a new sample by dot product of original sample and the coefficient vector $[[ a_{1} * [x_{1}], a_{2} * [x_{2}], a_{3} * [x_{3}], a_{4} * [x_{4}], a_{5} * [x_{5}] ]], [ a_{1} *[y_{1}], a_{2} * [y_{2}], a_{3} * [y_{3}], a_{4} * [y_{4}], a_{5} * [y_{5}] ]]$ with the shape [1,2,5] The classifier is fed by this new sample and outputs a logit vector $[logit_{1}, {logit_{2}, logit_{3}, logit_{4}, logit_{5} ]$ with the shape (5,). We take sigmoid (the maximum value in logit vector) as the possibility of a link existing between these two neurons.
>
>    In the SI of the revision, we will add a detail description about the architectures of models, encompassing explanations of their input and output.
>
>
> To Questions:
> 1. Fig. 4. For the continuously coupled models which we consider, the parent-child relationship does not correspond to a pre-post relationship. We agree that Fig. 4(c) is regrettably confusing, and in our revision we will update it with a pair of variables with more intuitive traces.
>
> 2. Connectivity of ANNs. In our response to Weakness 2 we explained these decisions, and we will also clarify these details in our revision.
>
> 3.	TENE does indeed refer to using Equation 5. In our revision, we state in Sec.4.1, “Transfer Entropy Neural Estimator (TENE), as in Eq. 5.” Also, for clarity, in Eq.5 we will change “TE(X→Y)” to “TENE(X→Y)”.

---

### Decision · Program_Chairs · 2023-09-21

**Decision:**

Accept (spotlight)

**Comment:**

The reviewers unanimously voted to accept the paper. The paper presents a method to identify critical regions of time series data where the coupling effect may manifest. To quote reviewer, f3Wr, "This paper provides convincing empirical evidence that the attentive transfer entropy is particularly useful in understanding coupling effects of realistic biological networks. They achieve this through first conducting ablation study on toy models of simple connectivity and comparing the proposed attentive transfer entropy with multiple commonly used baselines". I agree with the reviewer -- the paper is a good contribution and should be presented either as a spotlight or an oral.